# Deliberated Domain Bridging for Domain Adaptive Semantic Segmentation

**Lin Chen**[1,2]* **Zhixiang Wei**[1]* **Xin Jin**[3]* **Huaian Chen**[1]† **Miao Zheng**[2] **Kai Chen**[2] **Yi Jin**[1]†

[1] University of Science and Technology of China
[2] Shanghai AI Laboratory   [3] Eastern Institute for Advanced Study

{chlin, zhixiangwei}@mail.ustc.edu.cn, jinxin@eias.ac.cn, anchen@mail.ustc.edu.cn
{zhengmiao, chenkai}@pjlab.org.cn, jinyi08@ustc.edu.cn

## Abstract

In unsupervised domain adaptation (UDA), directly adapting from the source to the target domain usually suffers significant discrepancies and leads to insufficient alignment. Thus, many UDA works attempt to vanish the domain gap gradually and softly via various intermediate spaces, dubbed domain bridging (DB). However, for dense prediction tasks such as domain adaptive semantic segmentation (DASS), existing solutions have mostly relied on rough style transfer and how to elegantly bridge domains is still under-explored. In this work, we resort to data mixing to establish a *deliberated domain bridging (DDB)* for DASS, through which the joint distributions of source and target domains are aligned and interacted with each in the intermediate space. At the heart of DDB lies a *dual-path domain bridging* step for generating two intermediate domains using the coarse-wise and the fine-wise data mixing techniques, alongside a *cross-path knowledge distillation* step for taking two complementary models trained on generated intermediate samples as 'teachers' to develop a superior 'student' in a multi-teacher distillation manner. These two optimization steps work in an alternating way and reinforce each other to give rise to DDB with strong adaptation power. Extensive experiments on adaptive segmentation tasks with different settings demonstrate that our DDB significantly outperforms state-of-the-art methods. Code is available at https://github.com/xiaoachen98/DDB.git.

## 1 Introduction

When training deep models on one domain but applying it to other unseen domains, its performance typically drops seriously due to the domain shift/discrepancy issue [45, 44, 69, 55]. Since annotating data in the new scenario to re-train model to mitigate performance degradation is too expensive and time consuming, extensive researches have resorted to unsupervised domain adaptation (UDA) [41, 15, 34, 3], which aims to transfer knowledge from labeled source domain to unlabeled target domain.

Generally, existing UDA methods typically reduce the domain discrepancy by leveraging information statistics metrics [10, 27, 29, 33, 35, 46, 66] or adversarial training [15, 30, 34, 49, 54, 58, 3]. Both of these branches **directly** adapt the knowledge learned from the source domain to the target domain. However, excessive/continuous domain discrepancies tend to limit the efficiency of these methods for knowledge transfer, causing non-optimal performance, especially on dense prediction tasks such as semantic segmentation.

---

*Equal contribution.
†Corresponding author.

36th Conference on Neural Information Processing Systems (NeurIPS 2022).

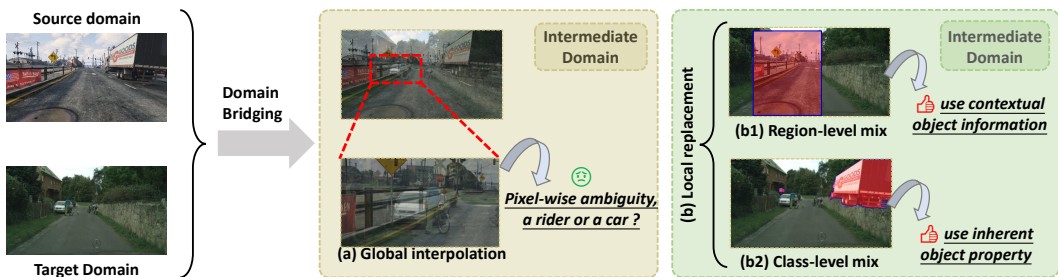

Figure 1: Different domain bridging ways for solving the domain-adaptive semantic segmentation (DASS) task. (a) the global interpolation based mix may cause an unexpected pixel-wise ambiguity issue; (b) the local replacement mix techniques better preserve the semantic consistency of pixels of the same object, which is important for segmentation. The coarse region-level local replacement helps the model exploit contextual information, while the fine class-level local replacement enables the model to exploit the inherent properties of each object, making them distinguishable.

To address this issue, some recent approaches tend to mitigate the excessive domain discrepancies by **gradually** transferring knowledge across domains by constructing intermediate domains in the input space [41, 60, 61], feature space [11, 12], or output space (self-training based) [64, 65]. Such mechanism of constructing intermediate domain is usually termed as domain bridging (DB). Despite unprecedented advances achieved in UDA classification task, the most common DB approaches, such as CycleGAN [72] and ColorTransfer [47] that based on style transfer, are inapplicable and cannot achieve satisfactory adaptation performance for the domain adaptive semantic segmentation (DASS). These style transfer-based DB approaches are prone to generate unexpected artifacts in the global input space and also ignore to bridge in the label space, which makes the optimization constraints insufficient for the dense prediction tasks such as DASS [16, 22, 67, 68, 71]. Therefore, such dilemma drives an urgent demand to investigate a new DB method for the densely predicted DASS area.

In this paper, to consistently construct intermediate representations in the input space as well as the label space for domain bridging in DASS, we resort to the mix-based data augmentation techniques[63, 62, 18, 14, 43]. First, we study the existing data mix methods and group them into two categories according to their working way: global interpolation [63] and local replacement [62, 18, 14, 43, 71]. As shown in Fig. 1, *the global interpolation based data mix may cause the pixel-wise ambiguity issue, but the local replacement based mix methods can better preserve the semantic integrity/consistency of objects for segmentation*. Next, to fully exploit the local replacement based data mix for domain bridging, we further deeply explore it from two complementary perspectives: the coarse region-level mix (*e.g.*, CutMix [62], FMix [18]) and the fine class-level mix (*e.g.*, ClassMix [43]). We can also see from Fig. 1 that the coarse region-level domain bridging helps the model to exploit contextual information, reducing semantics confusion (*e.g.*, category confusion between objects such as 'truck, bus, and train'). Complementarily, the fine class-level domain bridging enables the model to fully exploit the inherent properties of each category, making each object distinguishable. However, these two groups of DB methods tend to drive the model to be overly dependent on contextual information or inherent properties, causing class bias and confusion in the target domain separately.

In this work, we propose a powerful DASS method called Deliberated Domain Bridging (DDB) to carefully take advantage of data mixing techniques and gradually transfer knowledge from the source domain to the target domain. As an optimization strategy, DDB consists of two alternating steps, *i.e.*, Dual-Path Domain Bridging (DPDB) and Cross-path Knowledge Distillation (CKD). In the first step, DPDB independently leverages the coarse region-level data mix and fine class-level data mix to construct two complementary bridging paths to train two expert teacher models, achieving dual-granularity domain bridging. In the second step, CKD uses two complementary teacher models to guide one identical student model on the target domain, achieving adaptive segmentation. These two optimization steps work in an alternating way, which allows the powerful teacher and student models to reinforce each other progressively based on the joint distributions of source and target domains. The main contributions of this paper are summarized as follows:

- To the best of our knowledge, this is the first work that provides a comprehensive analysis w.r.t the recent domain bridging techniques when directly applied to the task of domain-adaptive semantic segmentation (DASS).

- Based on the analysis, we propose an effective DASS method called Deliberated Domain Bridging (DDB), which consists of two alternating steps – Dual-path Domain Bridging (DPDB) and Cross-path Knowledge Distillation (CKD). These two optimization steps promote each other and progressively encourage two complementary teacher models and a superior student model (used for inference), achieving a win-win effect.
- We experimentally validate the superiority of our DDB not only in the single-source domain setting but also in the multi-source and multi-target domain settings and conclude that DDB outperforms previous methods tailored for each setting by a large margin.

## 2    Related Work

**Domain Adaptive Semantic Segmentation (DASS).** This task aims to improve the adaptation performance for the semantic segmentation model to avoid laborious pixel-wise annotation in new target scenarios. The recent DASS works can be mainly grouped into two categories: adversarial training based methods [23, 51–53, 37] and self-training based methods [24, 73, 70, 64, 65]. For the first branch, most works tend to learn domain-invariant representations based on a min-max adversarial optimization game, where a feature extractor is trained to fool a domain discriminator and thus helps to obtain aligned feature distributions [51–53, 37]. The second branch focuses on how to generate highly reliable pseudo labels for the target domain data for further model optimization, which drives many classic related techniques, such as confidence regularized pseudo label generation [73, 70] and category-aware pseudo label rectification [64, 65]. These two branches of the DASS task both *directly* adapt the knowledge learned from the source domain to the target domain. However, the large continuous domain discrepancies in DASS make such direct discrepancy minimization paradigms difficult, due to the fine-grained pixel-wise gap among different domains.

**Domain Bridging (DB).** Instead of *directly* transferring knowledge from the source domain to the target domain, some UDA works in the tasks of classification and person re-identification tend to *gradually* transfer knowledge by building a bridge between source and target domains, *i.e.*, constructing an intermediate domain on the image level [57, 41], on the feature level [11, 12], or on the output level [64, 65]. Representatively, GVB [11] designs a gradually vanishing bridge and inserts it into the task-specific classifier and the domain discriminator to construct intermediate domain-invariant representations, reducing the knowledge transfer difficulty. Along this road, some works [57, 60, 41, 12, 5, 50] resort to style transfer techniques [6, 22, 7, 16] and data mix techniques [63, 62, 43] for constructing various intermediate domains. However, the existing DB approaches have not yet been extensively investigated in DASS. In this paper, we first perform a comprehensive analysis w.r.t the recent DB techniques and find the complementarity between the coarse region-level DB and the fine class-level DB methods, then deliberately/carefully apply these two DB methods to help the task of DASS.

## 3    Deliberated Domain Bridging

### 3.1    Recap of Preliminary Knowledge

For domain adaptive semantic segmentation (DASS), we denote the source domain as $D_s = \{(x_s^{(i)}, y_s^{(i)})\}_{i=1}^{N^s}$ with $N^s$ samples drawn from the source domain $\mathcal{S}$, where $x_s^{(i)} \in X_s$ is an image, $y_s^{(i)} \in Y_s$ is the corresponding pixel-wise one-hot label covering $K$ classes. Similarly, the unlabeled target domain set is denoted as $D_t = \{x_t^{(i)}\}_{i=1}^{N^t}$ with $N^t$ samples drawn from the target domain $\mathcal{T}$. Note that the source and target domains share the same label space. This work aims to learn a segmentation model for effectively transferring knowledge from the source domain to the target domain, finally achieving reliable pixel-wise predictions on the target data. Following previous works [64, 65], this segmentation model $M$ consists of a feature extractor that maps the image to the feature space and a classifier that generates corresponding pixel-wise predictions.

### 3.2    Exploring Domain Bridging for DASS

**Revisiting Existing DB Methods.** As mentioned in Sec. 2, the previous DB methods are mainly based on style transfer [72, 47], global interpolation based mix [63], and local replacement based mix [62, 18, 14, 43, 71]. Formally, the image-level style transfer-based DB methods can be formulated

Table 1: Performance (mIoU) comparison of different DB methods on GTA5→Cityscapes. S→T denotes that we translate the image from the source domain to the target domain. Pseudo Labeling represents the constructed self-training baseline without any DB method. + indicates that both are used, while ⊕ indicates that these methods will be used mutually with an equal probability. The best score is indicated in **underlined bold**.

(a) Comparison of style transfer-based DB methods.

| Method | mIoU |
|---|---|
| Source only | 26.3±0.9 |
| + CycleGAN [72] (S→T) | 37.8±0.4 |
| + Color Transfer [47] (S→T) | 38.7±1.2 |
| + FDA [61] (S→T) | 41.3±0.6 |
| Pseudo Labeling | 30.7±0.4 |
| + CycleGAN (T→S) | 28.9±0.5 |
| + Color Transfer (T→S) | 31.4±0.5 |
| + FDA (T→S) | **42.6**±0.6 |

(b) Comparison of global blending-based and region-based DB methods.

| Method | mIoU |
|---|---|
| Pseudo Labeling | 30.7±0.4 |
| + Mixup [63] | 31.6±0.6 |
| + CowMix [14] | 50.7±0.4 |
| + FMix [18] | 50.0±0.2 |
| + CutMix [62] | **54.9**±0.2 |
| + ClassMix [43] | 54.3±1.4 |

(c) Comparison of combined DB methods of different groups.

| Method | mIoU |
|---|---|
| Pseudo Labeling | 30.7±0.4 |
| + CutMix + CycleGAN (S→T) | 47.6±1.1 |
| + ClassMix + CycleGAN (S→T) | 53.9±1.0 |
| + CutMix + FDA (S→T) | 46.8±1.2 |
| + ClassMix + FDA (S→T) | 50.6±1.1 |
| + CowMix ⊕ CutMix | 51.7±0.4 |
| + FMix ⊕ CutMix | 50.6±0.7 |
| + FMix ⊕ ClassMix | 54.5±0.5 |
| + CutMix ⊕ ClassMix | **55.2**±1.0 |

as,

$$x_{s \to t} = h\left(x_s\right), \quad x_{t \to s} = h'\left(x_t\right), \tag{1}$$

where $h\left(\cdot\right)$ and $h'\left(\cdot\right)$ represent the $\mathcal{S} \to \mathcal{T}$ translation function and $\mathcal{T} \to \mathcal{S}$ translation function, respectively. Such style transfer-based DB approaches tend to generate unexpected artifacts at the image level and also ignore the influence of pixel-wise label correspondence.

In addition, we formalize the global interpolation mix based DB methods as,

$$\begin{aligned} x_{new} &= \lambda \cdot x_s + (1 - \lambda) \cdot x_t \\ y_{new} &= \lambda \cdot y_s + (1 - \lambda) \cdot y_t, \end{aligned} \tag{2}$$

where $\lambda$ denotes the mixing ratio sampled from a beta distribution. Furthermore, the local replacement mix based DB methods are formulated as,

$$\begin{aligned} x_{new} &= \mathbf{M} \odot x_s + (\mathbf{1} - \mathbf{M}) \odot x_t \\ y_{new} &= \mathbf{M} \odot y_s + (\mathbf{1} - \mathbf{M}) \odot y_t, \end{aligned} \tag{3}$$

where $\mathbf{M}$ denotes a binary mask indicating which pixel needs to be copied from the source domain and pasted to the target domain, $\mathbf{1}$ is a mask filled with ones, and $\odot$ represents the element-wise multiplication operation. $y_t$ represents the pseudo label for target domain. In particular, this local replacement DB contains two types of coarse region-level mix and fine class-level mix. As shown in Fig. 1(b), the binary mask $\mathbf{M}$ of the former is the cut patch [62, 18, 14] while $\mathbf{M}$ of the latter is obtained from the pixel-wise annotations in source domain [43].

**Analyzing DB Methods with Toy Game.** From the above formalization, we can see that the global interpolation-based and local replacement-based DB methods both build bridges across the cross-domain joint distributions of input data, which can benefit the densely predicted DASS task. To verify this, we perform a toy game with a simple self-training based DASS pipeline following [50, 24] to evaluate the performance w.r.t semantic segmentation of different DB methods. As illustrated in Tab. 1 (a) and (b), although style transfer based and global interpolation based DB methods both outperform baseline (*i.e.*, the source only scheme), they are pronouncedly inferior to their local replacement-based counterparts. This implies that for the DASS task, (1) it not only needs to construct an intermediate domain on the input space, but also the label space; (2) the local replacement based DB methods are more suitable for segmentation because they can better preserve the semantic integrity/consistency for pixels belonging to the same object.

In addition, Tab. 1 (b) shows that the performance of coarse region-level CutMix [62] and fine class-level ClassMix [43] are comparable. Thus, we further conduct a group of tests by combining different DB methods for a deeper study. The results are shown in Tab. 1 (c), we can observe that (1) due to the unexpected artifacts, the segmentation performance is degraded when region-level DB methods are combined with style transfer ones; (2) we surprisingly notice that the coarse region-level and fine class-level domain bridging methods can mutually reinforce/promote each other.

**Analyzing the Local Replacement Based DB Methods with Visualization.** For the coarse region-level data mix methods (*e.g.*, CutMix [62]), those pixels (*i.e.*, a patch) pasted to the target domain

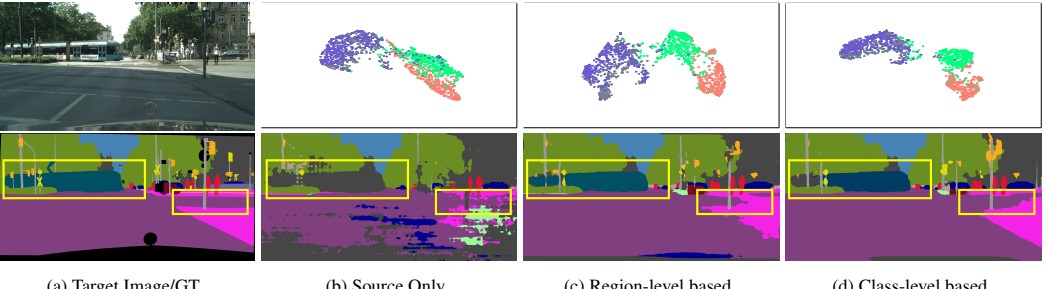

| (a) Target Image/GT | (b) Source Only | (c) Region-level based | (d) Class-level based |

Figure 2: Visualization of the qualitative results on GTA5 → Cityscapes benchmark (bottom row), and feature space on the val set utilizing UMAP [39] (top row). For a clear analysis of the feature space, we select two pairs of categories suffering class bias and confusion, *i.e.*, salmon for road, green for sidewalk, gray for bus, and purple for train.

usually have a rich contextual information. For example, the pixel beneath a 'person' must belong to the 'sidewalk'. However, the area beneath a 'person' is more likely to be 'road' in the target domain. Thus, such excessive *hard* reliance on context may introduce the issue of class bias from the source domain to the target domain, *e.g.*, 'road' and 'sidewalk' in Fig. 2 (c). In contrast, the fine class-level bridging method (*e.g.*, ClassMix [43]) only pastes a set of pixels belonging to the same class to the target domain image, avoiding class bias issue, which also drives the model to discriminate different objects solely based on their inherent properties, leading to a more compact feature distribution in Fig. 2 (d). But, the scheme that only relies on the characteristics of each category may be confused by the classes that can easily be distinguished by the context, *e.g.*, 'train' and 'bus' in Fig. 2 (d). All in all, both coarse-grained and fine-grained DB methods have their own advantages and drawbacks, and thus there is an urgent need to find a appropriate way to combine them to achieve a win-win effect.

### 3.3 Progressively learning from Dual-grained Domain Bridging

Instead of directly combining individual DB methods as did in the bottom of Tab. 1(c), we propose an alternating optimization strategy to progressively transfer knowledge from the source domain to the target domain, which consists of two steps, *i.e.*, Dual-Path Domain Bridging (DPDB) and Cross-path Knowledge Distillation (CKD). These two steps are conducted iteratively, and the ending of each round will serve as the beginning for the next round (see detailed algorithm in supplemental material).

**Dual-Path Domain Bridging (DPDB).** To better preserve and exploit the advantages of the coarse region-level and fine class-level DB methods, we create bridging paths for them independently rather than simply fusing them. Based on previous analysis and experiments, we utilize the cross-domain CutMix [62] and ClassMix [43] techniques to construct the coarse region-path (CRP) and fine class-path (FCP) domain bridging, respectively. The self-training pipeline then proceeds in the following manner along each path (here we take the coarse region-path (CRP) as an example for illustration):

Following [2], to minimize the empirical risk on the unlabeled target domain, we simultaneously minimize the empirical risk on the source domain and mitigate the domain discrepancy. The first item is achieved with a pixel-wise cross-entropy (CE) loss,

$$\mathcal{L}_{src}^{C} = - \sum_{i=1}^{H \times W} \sum_{j=1}^{K} y_s^{(i,j)} \log M_C(x_s)^{(i,j)}, \tag{4}$$

where $M_C$ is the model training on the coarse region-path and $H, W$ denote the sample height and width, respectively. To mitigate the domain discrepancy, we minimize the CE loss on the constructed bridging path instead of adversarial training or minimize the predefined discrepancy metric. Considering the fact that the unlabeled target domain images are involved in the bridging path construction, an additional teacher network $M_C'$ is employed to generate a denoised pixel-wise pseudo-label $\hat{y}_t$ for $x_t$ through the exponential moving average (EMA) based on the weights of $M_C$ after each training step $t$,

$$\theta_{M_C'}^{t+1} \leftarrow \alpha \theta_{M_C'}^{t} + (1 - \alpha) \theta_{M_C}^{t}, \tag{5}$$

where $\alpha$ denotes the momentum and is set to 0.99. Based on Eqn. 3, we can generate the bridging image $x_{crp}$ and label $\hat{y}_{crp}$ on the coarse region-path. In line with [50, 43], a confidence-based weight

map $m_{crp}$ will be generated to regularize the target domain during the training process as follows,

$$m_{crp} = \mathbf{M} \odot \mathbf{1} + (\mathbf{1} - \mathbf{M}) \odot m_t, \tag{6}$$

where $m_t = \frac{\sum_{i=1}^{H \times W} [\max_{j'} \mathcal{M}'_C(x_t)^{(i,j')} > \tau]}{H \cdot W}$ denotes the ratio of pixels that exceed a threshold $\tau$ on the maximum softmax probability and $[\cdot]$ represents the Iverson bracket. Then, we minimize the CE loss on the region-path bridging,

$$\mathcal{L}_{brg}^C = -\sum_{i=1}^{H \times W} \sum_{j=1}^{K} m_{crp}^{(i,j)} \hat{y}_{crp}^{(i,j)} \log M_C(x_{crp})^{(i,j)}. \tag{7}$$

Furthermore, the overall objective function of the self-training pipeline on the region-path bridging is summarized as,

$$\mathcal{L}_C = \mathcal{L}_{src}^C + \mathcal{L}_{brg}^C = \mathcal{L}_{ce}\left(M_C\left(x_s\right), y_s\right) + \mathcal{L}_{ce}\left(M_C\left(x_{crp}\right), m_{crp} \odot \hat{y}_{crp}\right). \tag{8}$$

Similarly, we can obtain the overall objective function $\mathcal{L}_F$ on the fine class-path (FCP) bridging. By minimizing $\mathcal{L}_C$ and $\mathcal{L}_F$ separately for the coarse region-path and fine class-path, we can obtain two complementary models. The next problem that needs to be addressed is how to appropriately integrate these two kinds of complementary knowledge in an elegant manner.

**Cross-path Knowledge Distillation (CKD).** Inspired by previous works [17, 21], knowledge can be transferred from a teacher network to a student network by knowledge distillation in the output space. Here, we reform it to extract knowledge from two complementary teachers and adaptively transfer the integrated knowledge to a student. Note that we only integrate and transfer the complementary knowledge in the unlabeled target domain. Specifically, the outputs of two teachers have been adaptively weighted and ensembled as guidance to drive the student model to learn segmentation in the unlabeled target domain. Furthermore, we experimentally choose the 'hard' distillation, *i.e.*, ensembling the predicted softmax logits to generate a one-hot vector and utilizing the CE loss for supervising the student model $M_S$. The detailed distillation loss can be written as,

$$\mathcal{L}_{distill} = \mathcal{L}_{ce}\left(M_S\left(x_t^{aug}\right), \bar{y}_t\right), \tag{9}$$

where $x_t^{aug}$ represents the target images augmented by color jitter and gaussian blur, and $\bar{y}_t$ is obtained by weighted ensembling on the teachers' softmax logits of the target image $x_t$. Intuitively, different samples, even different pixels in one sample, require different contributions from the two teacher models for ensembling. We adaptively generate a pixel-wise weight map $w^{(i,j)}$ in the target domain by calculating the distance pixel-by-pixel between feature response $f^{(i)}$ before the classification layer and the centroid $\eta^{(j)}$ of each category. At each location, the closer the feature response is far from one centroid of certain category, the more likely it belongs to that category and thus should contribute more to the ensemble effect. Therefore, taking the coarse region-path (CRP) as an example, we first utilize the trained $M_C$ to calculate the centroid $\eta_C^{(j)}$ of each category in the target domain,

$$\eta_C^{(j)} = \frac{\sum_{x_t \in X_t} \sum_i f_C^{(i)} * \mathbb{1}(\hat{y}_t^{(i,j)} == 1)}{\sum_{x_t \in X_t} \sum_i \mathbb{1}(\hat{y}_t^{(i,j)} == 1)}. \tag{10}$$

Then, we define the adaptive ensemble weights $w_C^{(i,j)}$ of $M_C$ as the softmax over feature distances to the centroids

$$w_C^{(i,j)} = \frac{\exp(-\|f_C^{(i)} - \eta_C^{(j)}\|)}{\sum_{j'} \exp(-\|f_C^{(i)} - \eta_C^{(j')}\|)}. \tag{11}$$

Similarly, we can also obtain the ensembling weight of the other path $w_F^{(i,j)}$ of $M_F$. Furthermore, we can obtain the pseudo-label $\bar{y}_t$, following the weighted ensembling,

$$\bar{y}_t = \arg\max\left(\frac{w_C \cdot \sigma(M_C(x_t)) + w_F \cdot \sigma(M_F(x_t))}{2}\right), \tag{12}$$

where $\sigma$ denotes the softmax function. In addition, the student model is also supervised by the labeled source data to generate discriminative features

$$\mathcal{L}_{src}^S = \mathcal{L}_{ce}(M_S(x_s), y_s). \tag{13}$$

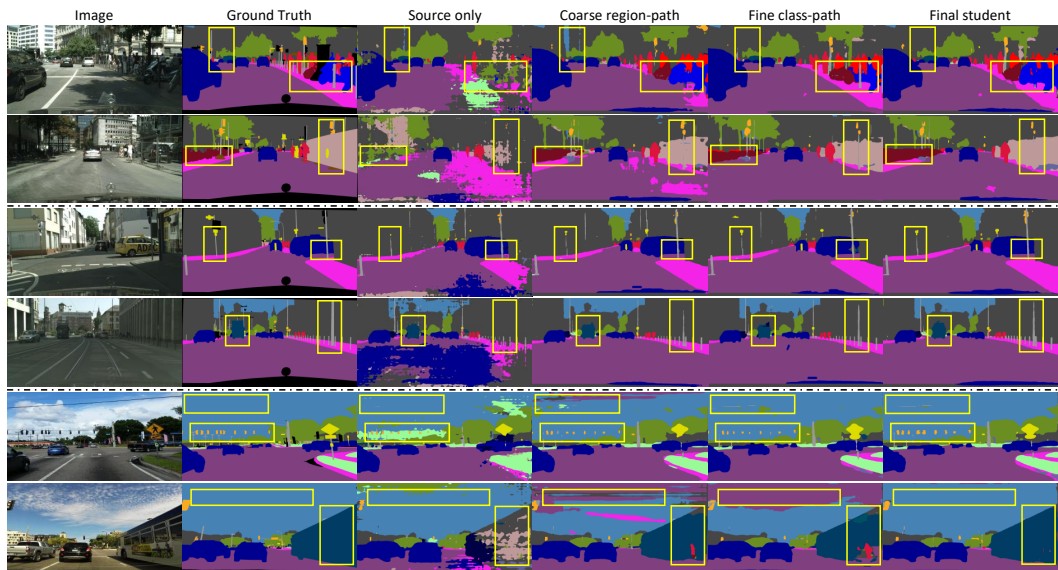

Figure 3: Comparison of qualitative results on GTA5 → Cityscapes (top area), GTA5 +Synscapes → Cityscapes (middle area), and GTA5 → Cityscapes + Mapillary (bottom area) benchmarks.

Table 2: Comparison results of GTA5→Cityscapes adaptation (using ResNet-101 as the backbone and DeepLabv2 as the head) in terms of mIoU. *distill* denotes applying multi-rounds self-distillation for the student network initialized by self-supervised pre-training. The best score is indicated in **underlined bold**.

| Method | road | sidewalk | building | wall | fence | pole | light | sign | vege. | terrain | sky | person | rider | car | truck | bus | train | motor | bike | mIoU |
|---|---|---|---|---|---|---|---|---|---|---|---|---|---|---|---|---|---|---|---|---|
| Source only | 75.8 | 16.8 | 77.2 | 12.5 | 21.0 | 25.5 | 30.1 | 20.1 | 81.3 | 24.6 | 70.3 | 53.8 | 26.4 | 49.9 | 17.2 | 25.9 | 6.5 | 25.3 | 36.0 | 36.6 |
| CyCADA [22] | 86.7 | 35.6 | 80.1 | 19.8 | 17.5 | 38.0 | 39.9 | 41.5 | 82.7 | 27.9 | 73.6 | 64.9 | 19.0 | 65.0 | 12.0 | 28.6 | 4.5 | 31.1 | 42.0 | 42.7 |
| ADVENT [52] | 89.4 | 33.1 | 81.0 | 26.6 | 26.8 | 27.2 | 33.5 | 24.7 | 83.9 | 36.7 | 78.8 | 58.7 | 30.5 | 84.8 | 38.5 | 44.5 | 1.7 | 31.6 | 32.4 | 45.5 |
| BDL [31] | 91.0 | 44.7 | 84.2 | 34.6 | 27.6 | 30.2 | 36.0 | 36.0 | 85.0 | 43.6 | 83.0 | 58.6 | 31.6 | 83.3 | 35.3 | 49.7 | 3.3 | 28.8 | 35.6 | 48.5 |
| FADA [53] | 91.0 | 50.6 | 86.0 | 43.4 | 29.8 | 36.8 | 43.4 | 25.0 | 86.8 | 38.3 | 87.4 | 64.0 | 38.0 | 85.2 | 31.6 | 46.1 | 6.5 | 25.4 | 37.1 | 50.1 |
| CAG [65] | 90.4 | 51.6 | 83.8 | 34.2 | 27.8 | 38.4 | 25.3 | 48.4 | 85.4 | 38.2 | 78.1 | 58.6 | 34.6 | 84.7 | 21.9 | 42.7 | 41.1 | 29.3 | 37.2 | 50.2 |
| IAST [40] | 93.8 | 57.8 | 85.1 | 39.5 | 26.7 | 26.2 | 43.1 | 34.7 | 84.9 | 32.9 | 88.0 | 62.6 | 29.0 | 87.3 | 39.2 | 49.6 | 23.2 | 34.7 | 39.6 | 51.5 |
| DACS [50] | 89.9 | 39.7 | 87.9 | 30.7 | 39.5 | 38.5 | 46.4 | 52.8 | 88.0 | 44.0 | 88.8 | 67.2 | 35.8 | 84.5 | 45.7 | 50.2 | 0.0 | 27.3 | 34.0 | 52.1 |
| SAC [1] | 90.4 | 53.9 | 86.6 | 42.4 | 27.3 | 45.1 | 48.5 | 42.7 | 87.4 | 40.1 | 86.1 | 67.5 | 29.7 | 88.5 | 49.1 | 54.6 | 9.8 | 26.6 | 45.3 | 53.8 |
| CTF [38] | 92.5 | 58.3 | 86.5 | 27.4 | 28.8 | 38.1 | 46.7 | 42.5 | 85.4 | 38.4 | **91.8** | 66.4 | 37.0 | 87.8 | 40.7 | 52.4 | **44.6** | 41.7 | 59.0 | 56.1 |
| ProDA [64] | 91.5 | 52.4 | 82.9 | 42.0 | 35.7 | 40.0 | 44.4 | 43.8 | 87.0 | 43.8 | 79.5 | 66.5 | 31.4 | 86.7 | 41.1 | 52.5 | 0.0 | 45.4 | 53.8 | 53.7 |
| ProDA+*distill* | 87.8 | 56.0 | 79.7 | **46.3** | 44.8 | 45.6 | 53.5 | 53.5 | 88.6 | 45.2 | 82.1 | 70.7 | 39.2 | 88.8 | 45.5 | 59.4 | 1.0 | 48.9 | 56.4 | 57.5 |
| UndoDA [32] | 89.1 | 34.3 | 83.6 | 38.3 | 27.5 | 28.9 | 34.7 | 17.6 | 84.2 | 41.0 | 85.1 | 57.8 | 33.7 | 85.1 | 38.5 | 41.3 | 30.7 | 31.1 | 48.0 | 49.0 |
| UndoDA+ProDA | 92.9 | 52.7 | 87.2 | 39.4 | 41.3 | 43.9 | 55.0 | 52.9 | **89.3** | **48.2** | 91.2 | 71.4 | 36.0 | 90.2 | **67.9** | 59.8 | 0.0 | 48.5 | 59.3 | 59.3 |
| CPSL [28] | 91.7 | 52.9 | 83.6 | 43.0 | 32.3 | 43.7 | 51.3 | 42.8 | 85.4 | 37.6 | 81.1 | 69.5 | 30.0 | 88.1 | 44.1 | 59.9 | 24.9 | 47.2 | 48.4 | 55.7 |
| CPSL+*distill* | 92.3 | 59.9 | 84.9 | 45.7 | 29.7 | **52.8** | **61.5** | **59.5** | 87.9 | 41.5 | 85.0 | **73.0** | 35.5 | 90.4 | 48.7 | **73.9** | 26.3 | **53.8** | 53.9 | 60.8 |
| Source only | 60.4 | 15.1 | 58.3 | 8.7 | 21.3 | 20.9 | 33.2 | 22.4 | 77.7 | 8.6 | 71.3 | 55.8 | 13.2 | 77.0 | 22.8 | 22.1 | 0.4 | 14.1 | 6.1 | 32.1 |
| DDB(Ours) | **95.3** | **67.4** | **89.3** | 44.4 | **45.7** | 38.7 | 54.7 | 55.7 | 88.1 | 40.7 | 90.7 | 70.7 | **43.1** | **92.2** | 60.8 | 67.6 | 34.2 | 48.7 | **63.7** | **62.7** |

The overall loss function of constraining the student model $M_S$ can be written as,

$$\mathcal{L}_{\mathcal{S}} = \mathcal{L}_{src}^{S} + \mathcal{L}_{distill} = \mathcal{L}_{ce}(M_S(x_s), y_s) + \mathcal{L}_{ce}(M_S(x_t^{aug}), \bar{y}_t). \tag{14}$$

In the end, a superior student model is well trained by adaptively integrating the knowledge from two complementary teacher models covering different granularities.

**Alternating Optimization Strategy.** By integrating the complementary knowledge from two expert teacher models, we can obtain a superior student model. In turn, this student model can be used to initialize the teacher models in the next new round, resulting in two stronger teacher models. They promote each other, achieving a win-win effect. Ultimately, we can obtain the most powerful student model after the final round.

Table 3: Comparison results of GTA5 (G) + Synscapes (S) → Cityscapes (C) adaptation (using ResNet-101 as the backbone and DeepLabv2 as the head) in terms of mIoU. The best score is indicated in **underlined bold**.

| Method | road | sidewalk | building | wall | fence | pole | light | sign | vege. | terrain | sky | person | rider | car | truck | bus | train | motor | bike | mIoU |
|---|---|---|---|---|---|---|---|---|---|---|---|---|---|---|---|---|---|---|---|---|
| Source only | 85.1 | 36.9 | 84.1 | 39.0 | 33.3 | 38.7 | 43.1 | 40.2 | 84.8 | 37.1 | 82.4 | 65.2 | 37.8 | 69.4 | 43.4 | 38.8 | 34.6 | 33.2 | 53.1 | 51.6 |
| AdaptSeg [51] | 89.3 | 47.3 | 83.6 | 40.3 | 27.8 | 39.0 | 44.2 | 42.5 | 86.7 | 45.5 | 84.5 | 63.1 | 38.0 | 79.4 | 34.9 | 48.3 | 42.1 | 30.7 | 52.3 | 53.7 |
| ADVENT [52] | 91.8 | 49.0 | 84.6 | 39.4 | 31.5 | 39.9 | 42.9 | 43.5 | 86.3 | 45.1 | 84.6 | 65.3 | 41.0 | 87.1 | 37.9 | 49.2 | 31.0 | 30.3 | 48.8 | 54.2 |
| MDAN [67] | 92.4 | 56.1 | 86.8 | 42.7 | 32.9 | 39.3 | 48.0 | 40.3 | 87.2 | 47.2 | 90.5 | 64.1 | 35.9 | 87.8 | 33.8 | 48.6 | 39.0 | 27.6 | 49.2 | 55.2 |
| MADAN [68] | 94.1 | 61.0 | 86.4 | 43.3 | 32.1 | 40.6 | 49.0 | 44.4 | 87.3 | 47.7 | 89.4 | 61.7 | 36.3 | 87.5 | 35.5 | 45.8 | 31.0 | 33.5 | 52.1 | 55.7 |
| MSCL [19] | 93.6 | 59.6 | 87.1 | 44.9 | 36.7 | 42.1 | 49.9 | 42.5 | 87.7 | 47.6 | 89.9 | 63.5 | 40.3 | 88.2 | 41.0 | 58.3 | 53.1 | 37.9 | 57.7 | 59.0 |
| Source only | 82.5 | 42.4 | 79.0 | 27.2 | 31.7 | 40.8 | 53.0 | 45.6 | 85.3 | 30.9 | 80.6 | 68.7 | 35.7 | 78.3 | 39.0 | 42.7 | 9.6 | 37.3 | 55.9 | 50.9 |
| DDB(Ours) | **96.9** | **75.6** | **90.0** | **54.4** | **48.6** | **47.6** | **61.1** | **66.3** | **89.7** | **48.4** | **93.4** | **74.4** | **52.7** | **92.3** | **60.8** | **74.7** | **58.9** | **53.9** | **71.4** | **69.0** |

Table 4: Comparison results of GTA5 (G) → Cityscapes (C) + Mapillary (M) adaptation (using ResNet-101 as the backbone and DeepLabv2 as the head) in terms of mIoU. The best score is indicated in **underlined bold**.

| Method | Target | road | sidewalk | building | wall | fence | pole | light | sign | vege. | terrain | sky | person | rider | car | truck | bus | train | motor | bike | mIoU | Avg. |
|---|---|---|---|---|---|---|---|---|---|---|---|---|---|---|---|---|---|---|---|---|---|---|
| Source only | C | 53.3 | 15.2 | 56.6 | 8.2 | 26.2 | 21.2 | 30.7 | 22.2 | 76.3 | 9.3 | 53.3 | 55.3 | 15.5 | 72.9 | 21.5 | 4.9 | 0.9 | 20.2 | 7.4 | 30.1 | 32.8 |
|  | M | 55.7 | 27.1 | 55.3 | 9.9 | 20.6 | 22.7 | 33.3 | 31.6 | 68.4 | 21.1 | 70.6 | 53.5 | 30.9 | 72.7 | 32.3 | 11.6 | 5.6 | 36.3 | 14.9 | 35.5 |  |
| CCL [25] | C | - | - | - | - | - | - | - | - | - | - | - | - | - | - | - | - | - | - | - | 45.1 | 46.8 |
|  | M | - | - | - | - | - | - | - | - | - | - | - | - | - | - | - | - | - | - | - | 48.8 |  |
| ADAS [26] | C | 88.3 | 32.2 | 82.2 | 23.8 | 24.2 | 30.5 | 35.0 | 33.3 | 83.3 | 37.9 | 85.1 | 56.7 | 21.9 | **84.6** | 38.6 | 46.2 | 0.5 | 33.5 | 33.3 | 45.8 | 47.5 |
|  | M | 84.2 | 33.9 | 78.5 | 25.5 | 24.5 | **35.6** | 39.8 | **52.4** | **71.2** | 40.2 | **92.4** | 58.7 | 38.7 | 82.7 | 44.4 | 46.4 | 15.2 | 37.8 | 32.2 | 49.2 |  |
| DDB(Ours) | C | **93.5** | **67.8** | **88.3** | **38.4** | **45.6** | **32.3** | **54.2** | **57.9** | **89.2** | **48.6** | **91.6** | **69.1** | **43.2** | **84.6** | **63.6** | **61.8** | **15.1** | **44.1** | **58.6** | **60.4** | **58.6** |
|  | M | **89.3** | **60.8** | **81.4** | **35.9** | **38.4** | 32.9 | **48.5** | 50.5 | 69.9 | **37.9** | 90.1 | **62.6** | **49.6** | **86.0** | **62.7** | **62.9** | **26.1** | **52.0** | **42.8** | **56.9** |  |

# 4 Experiments

## 4.1 Experimental Settings

**Datasets:** We use four publicly available semantic segmentation benchmarks for validation, including two synthetic scenes and two real-world scenes. Each scene has a unique structure and visual appearance. In detail, GTA5 [48] is a synthetic dataset of 24,966 labeled images obtained from a video game. Synscapes [56] is also a synthetic dataset of 25,000 images created by photo-realistic rendering techniques, and its style is closer to real-world driving scenes than GTA5. Cityscapes [9] is a real-world urban dataset collected from European cities, with 2,975 images for training and 500 images for validation. Mapillary Vista [42] is a large-scale dataset collected by various imaging devices worldwide and includes 18,000 images for training and 2,000 images for validation.

**Implementation details:** We use the mmsegmentation [8] codebase and train models on RTX 3090Ti GPUs. Following previous works [64, 65, 24, 59], we use the advanced DeepLab-v2 [4] model with ResNet101 [20] pre-trained on ImageNet-1k [13] as backbone, and train the model with AdamW [36]. We set the learning rate as 6e-5 for the backbone and 6e-4 for the decoder head, use a weight decay of 0.01 and a linear learning rate warmup followed by 1.5k iterations linear decay. All experiments are trained on a batch of 512x512 random cropped images for 40k iterations. We set the batch size to 2 for analysis and experiments in Tab. 1 and Tab. 6, and set batch size to 4 for other results. Following [50], we use the same augmentation parameters and set $\tau = 0.968$. For CutMix [62], the ratio of the selected region for cross-domain pasting is experimentally set to 0.3. For ClassMix [43], half of categories in the source domain are selected for cross-domain pasting.

## 4.2 Comparison with State-of-the-arts under Multiple Settings

**GTA5 → Cityscapes (single-source).** Tab. 2 reports results on the validation set of Cityscapes. Note that the comparable ProDA [64], UndoDA [32], and CPSL [28] have been improved with a warmup stage following existing DASS methods [51, 53]. Additionally, they also need to complete numerous rounds of self-distillation for the student model initialized by self-supervised pre-training. Compared to these methods that require redundant optimization processes, the proposed DDB method requires only two alternating optimization steps of DPDB and CKD. Despite simplicity, our method achieves a SOTA mIoU score of 62.7, outperforming existing methods significantly, which also achieves the best IoU score in 7 out of 19 categories. Particularly, thanks to combining the complementary teacher

Table 5: Ablation studies on the key components of our proposed method on the GTA5 (G) → Cityscapes (C) benchmark. It represents using the 'soft distillation' if the 'hard distillation' column isn't ticked.

| | components | | mIoU | gain |
|---|---|---|---|---|
| | source only | | 32.1 | |

| stage 1 DPDB | region path | class path | mIoU | gain |
|---|---|---|---|---|
| | ✓ | | 56.5 | +24.4 |
| | | ✓ | 58.2 | +26.1 |

| stage 1 CKD | hard distillation | adaptive ensemble | mIoU | gain |
|---|---|---|---|---|
| | ✓ | | 59.0 | +26.9 |
| | | | 59.9 | +27.8 |
| | | ✓ | 61.1 | +29.0 |
| | ✓ | ✓ | 61.2 | +29.1 |

| stage 2 DPDB | region path | class path | mIoU | gain |
|---|---|---|---|---|
| | ✓ | | 61.4 | +29.3 |
| | | ✓ | 62.6 | +30.5 |

| stage 2 CKD | hard distillation | adaptive ensemble | mIoU | gain |
|---|---|---|---|---|
| | ✓ | ✓ | 62.7 | +30.6 |

Table 6: Comparison results of ensemble from single and distinct paths in terms of mIoU on GTA5 (G) → Cityscapes (C). $\mathcal{M}_C^1$ and $\mathcal{M}_F^1$ denote the first run for the coarse region-path and fine class-path model, separately. The best score is indicated in **underlined bold**. And the cross-path ensemble is indicated in *italic*.

| | $\mathcal{M}_F^1$ | $\mathcal{M}_F^2$ | $\mathcal{M}_C^1$ | $\mathcal{M}_C^2$ |
|---|---|---|---|---|
| $\mathcal{M}_F^1$ | 55.5 | - | - | - |
| $\mathcal{M}_F^2$ | 55.9 | 55.8 | - | - |
| $\mathcal{M}_C^1$ | *56.4* | ***56.8*** | 55.0 | - |
| $\mathcal{M}_C^2$ | *56.2* | *56.6* | 55.7 | 54.7 |

Table 7: Performance comparison (mIoU) of the student model obtained from CKD during various rounds in the whole training process on three benchmarks. The best score is indicated in **underlined bold**.

| | 0 | 1 | 2 | 3 |
|---|---|---|---|---|
| G → C | 32.1 | 61.2 | **62.7** | **62.7** |
| G + S → C | 50.9 | 68.6 | **69.0** | 68.8 |
| G → C + M | 32.8 | 57.4 | **58.6** | 58.2 |

models devoted to exploiting the context and inherent properties by the coarse region-path and fine class-path, the student model performs surprisingly well in those categories that are susceptible to the class bias issue, *e.g.*, 'sidewalk' and 'bike.' Additionally, the final student model also performs well in those categories suffering from semantic confusion, *e.g.*, 'person, rider' and 'truck, bus, train.'

**GTA5 + Synscapes → Cityscapes (multi-source).** We also perform experiments under the multi-source domain setting. As shown in Tab. 3, our method obtains an impressive performance of 69.0 in mIoU, outperforming the previous SOTA methods over 10.0, achieving the best performance in all classes, especially those suffering class bias issue, *e.g.*, 'sidewalk' and 'bike.' Although the proposed DDB is not tailored for multi-source DASS, our method still benefits this task by constructing intermediate domains between target and multiple source domains to facilitate knowledge transfer.

**GTA5 → Cityscapes + Mapillary (multi-target).** Tab. 4 displays the performance of the proposed method in a multi-target domain setting. Although the multi-target DASS is more challenging due to unknown distributions, our method can still achieve an impressive performance of 58.6 in mIoU on average for multiple target domains, outperforming the existing SOTA methods by a significant margin. Moreover, the proposed DDB outperforms ADAS [26] by 31.2 in averaged mIoU on the 'sidewalk,' by 14.1 in averaged mIoU on the 'bus, train,' which indicates our method avoids the class bias and confusion issues. Such substantial performance gains comes from the DPDB-driven complementary teacher networks and the CKD-driven knowledge integration.

### 4.3 Ablation Study and Detailed Discussion

**Complementarity Verification.** To verify the complementarity of two teacher models trained on different bridging paths, we conduct an ablation where we train each path twice separately to obtain four different models. After ensembling these models pair-by-pair, the segmentation results are presented in Tab. 6. Unsurprisingly, the ensembled models across paths consistently perform better results than those only from a single view.

**Study on Dual-path Domain Bridging.** In Tab. 5, the source-only model achieves 32.1 in mIoU on the target domain. Combining the self-training pipeline with the coarse region-level and fine class-level domain bridging, we can obtain two complementary teacher models, and they achieve 56.5 and 58.2 in mIoU, respectively. As shown in Fig. 3, the coarse region-path tends to promote the model to utilize contextual information for prediction, whereas the fine class-path enables the model to focus more on exploiting inherent properties. Detailed results for the two teacher models in each category are provided in the **supplemental materials**.

**Study on Cross-path Knowledge Distillation.** Since CKD is performed on the unlabeled target domain, as shown in Tab. 5, we use the more stable hard distillation, which performs 0.9 mIoU higher than the soft distillation using the Kullback-Leibler divergence. Furthermore, our proposed adaptive ensemble scheme further improves the performance by 2.1 and 1.3 in mIoU in the case of soft and hard distillation, respectively. After applying CKD equipped with the hard distillation and adaptive

ensemble schemes, we can consistently obtain a superior student model. Fig. 3 illustrates how the student model performs after integrating the knowledge from two complementary teacher models and alleviates the class bias and confusion issues in various domain settings.

**Influence of Alternating Optimization Strategy.** As shown in Tab. 5, the alternation of DPDB and CKD allows the complementary teacher and student models to promote each other and gradually transfer knowledge across domains. We also test different alternating rounds on all three benchmarks, and report the performance of the student models after each round in Tab. 7 (more results are provided in the **supplemental materials**). The student model performs best across all three domain settings in the second round. On the other hand, the student model shows a slight performance degradation after the third round of alternate training in the multi-source and multi-target domain settings. We analyse the degradation is because the non-negligible domain conflict in these two settings.

## 5 Conclusions

In this paper, we study how the domain bridging techniques should be applied to domain adaptive semantic segmentation. To ensure that the segmentation model takes full advantage of domain bridging while avoiding side effects, we propose an effective Deliberated Domain Bridging (DDB) method. We build dual-path domain bridging (DPDB) with the coarse region-level data mix and fine class-level data mix to construct two complementary teacher models. Then, a superior student model can be generated from cross-path distillation (CKD) based on such two teacher models. By alternating steps of DPDB and CKD, teacher models and student model would promote each other and progressively transfer knowledge from the source domain to the target domain. Extensive ablation studies demonstrate the effectiveness of our method, and the experimental results on three benchmarks in the different settings further show its versatility and robustness.

## 6 Acknowledgement

This work was supported in part by the National Natural Science Foundation of China under Grant 61727809, in part by the Special Fund for Key Program of Science and Technology of Anhui Province under Grant 201903c08020002, and in part by the National Key Research and Development Program of China under Grant 2019YFC0117800.

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
