# OpenReview forum: "Deliberated Domain Bridging for Domain Adaptive Semantic Segmentation"
_NeurIPS.cc/2022/Conference — NeurIPS 2022 Accept_

### Official Review · Reviewer_PF8F · 2022-07-09

**Rating:** 6
**Confidence:** 4
**Soundness:** 2 fair
**Presentation:** 1 poor
**Contribution:** 3 good

**Summary:**

This paper is about unsupervised domain adaptation for the task of semantic segmentation. The paper argues for the importance of gradually bridging the domain gap, instead of a direct attempt to transfer a model from the source to the target domain. Motivated by an empirical analysis about different data mixing technologies, the paper explores two region-based mixing strategies (coarse regions and finer class-wise/mask regions) as domain bridges. Specifically, two models are trained on the two different domain bridges (Dual-Path-Domain-Bridge), which act as ensembled supervision for a single student model (Cross-path knowledge distillation). This student model can then initialize the teacher models for another round of these two steps. Experimental results in three different settings confirm the effectiveness of the proposed approach with state-of-the-art results on standard benchmarks.

**Questions:**

Please see few questions under "weaknesses" above

**Limitations:**

Potential societal impacts are not discussed. I think a paper on domain adaptation should include a discussion about potential biases that are carried over from source domains (specifically because these are often synthetic data which often contains some hand-crafted components, like object sampling distributions, etc.).

**Strengths And Weaknesses:**

### Strengths
- The ablation studies in Tables 5-7 are great and showcase the impact of individual components
- The results are impressive, with a clear improvement on the standard benchmark (GTA5->CityScapes), as well as other settings (multi-source and multi-target)
- The experiments in Table 1 are a good motivation for the choice of data mixing strategies (CutMix and ClassMix)



### Weaknesses

- The writing, specifically the motivation and positioning with respect to prior work, needs improvement.
  - There may be alternative domain bridges than data mixing, like methods that rely on self-training and choose confident target pseudo labels as intermediate source domain.
  - The justification for exploring new domain bridges in lines 37-41 is vague and unclear: What are "unexpected artifacts in the global input space"? What "optimization constraints" are referred to?
- I do not see why the paper only evaluates two domain bridging strategies in the ensemble. One could also include more. Relating to ensemble methods, one could expect improvements if an additional data mixing strategy is "orthogonal". One recent successful example for a global mixing strategy is [A] and could be easily integrated.
- There is a related work on domain bridges for semantic segmentation that was not included: [B]
- In line 185, shouldn't the reference go to Eq. 3?
- I do not quite understand why the mixing weights in Eq. 11/12 help. Aren't the softmax values (i.e., scores) already an indication how far away a sample is from the decision boundary?
- It would be good to point the reader to the supplemental material for a detailed description of the training strategy.
- It's hard to understand and see details in Figure 1.


**References:**
- [A] FDA: Fourier Domain Adaptation for Semantic Segmentation. Yang and Soatto. CVPR'20
- [B] Domain Bridge for Unpaired Image-to-Image Translation and Unsupervised Domain Adaptation. Pizzati et al. WACV'20

---

> ### Author Response · Authors · 2022-08-02
> **Response to Reviewer PF8F (Q5 to Q9)**
>
> **Q5: There is a related work on domain bridges for semantic segmentation that was not included: [B].**
>
> **A5:** As we have replied in Q1, this work of [B] falls under the category of generative style transfer DB methods and is not our focus. We have already cited it in the revision.
>
> **Q6: In line 185, shouldn't the reference go to Eq. 3?**
>
> **A6:** Thanks. We have already corrected it in the revision.
>
> **Q7: Why do the mixing weights in Eq. 11/12 help? Aren't the softmax values (i.e., scores) already an indication of how far away a sample is from the decision boundary?**
>
> **A7:** The parameterized softmax layer is more prone to be biased toward the supervised source domain than the pseudo-labeled target domain, which hinders the prediction in the target domain. Thus we compute the feature centroid for each class over the entire target domain and modulate the predicted probabilities on the fly according to the distance between the feature response and the feature centroids for the teacher network.
>
> **Q8: It would be good to point the reader to the supplemental material for a detailed description of the training strategy. And it's hard to understand and see the details in Figure 1.**
>
> **A8:** Thank you for the valuable suggestions. Figure 1 is just for explaining the motivation. Due to the limited space, we put the algorithm in supplementary material rather than a framework figure in the main paper. And we have pointed out the location of the algorithm at the start of Sec. 3.3 of the revision for a clearer understanding of the proposed framework.
>
> **Q9: Potential societal impacts are not discussed.**
>
> **A9:** We are sorry for the missing discussions about the potential societal impacts. It might be used in undesirable applications like surveillance or military UAVs for the purpose of domain adaptive semantic segmentation. Legal limitations on the applications of semantic segmentation algorithms could be a potential defense. And we have already updated this in the limitation section of the revision.

---

> ### Author Response · Authors · 2022-08-02
> **Response to Reviewer PF8F (Q1 to Q4)**
>
> Thanks for your time to carefully read our paper and give valuable suggestions. We have fully addressed your concerns below.
>
> **Q1: Clarification about the positioning with respect to prior work.**
>
> **A1:** Thanks for your reminder. We will add an extra explanation to illustrate the position of our work w.r.t prior works in the next revision: a comprehensive and broader formulation of conducting domain bridging (DB) could be divided into three branches – in the input space, in the output space, and the joint space. FDA[A] and the generative-based methods belong to the first branch that performs DB in the input space, self-training-based methods belong to the second one, and data mixing-based methods belong to the last one. The proposed framework mainly focuses on taking full advantage of complementary data-mixing-based methods and belongs to the last branch as well.
>
> **Q2: What does "unexpected artifacts in the global input space" mean? What "optimization constraints" are referred to?**
>
> **A2:** As shown in [15, 21, 66] and our re-implementation results, these DB methods based on generative style transfer techniques are prone to produce **unexpected artifacts** when transforming source domain images to target domain styles (**especially** in the case of **high-resolution** inputs), such as producing additional tree-covered areas in the sky and overly "imagining" palm trees as leafy trees (e.g., Figure 6(b) in [21], and Figure 3 in [66]). The **optimization constraints** arise from the fact that, regardless of the semantics of the artifact regions, their corresponding pixel-level Ground-Truth would remain unchanged.
>
> **Q3: Why the paper only evaluates two domain bridging strategies in the ensemble?**
>
> **A3:** As shown in Table 1(c) of the main paper, we have experimented with style transfer-based, global blending-based, coarse region-based, and fine region-based domain bridging (DB) methods individually or in combination and found that the combination of coarse region-based DB methods and fine region-based ones outperforms all others. Moreover, here we further conducted model ensemble experiments among the ClassMix, FMix, and CowMix methods, and that’s why we chose such two representative strategies of coarse-based and fine-based DBs for implementation.
>
> *Table A: The comparison of model ensemble experiments between FMix and ClassMix. The number (e.g., 1) denotes different runs of the same DB method.*
> |           | FMix1 | FMix2 | ClassMix1 | ClassMix2 |
> |-----------|-------|-------|-----------|-----------|
> | FMix1     | 49.9  | -     | -         | -         |
> | FMix2     | 50.8  | 50.2  | -         | -         |
> | ClassMix1 | 54.3  | 55.0  | 55.5      | -         |
> | ClassMix2 | 54.5  | 55.2  | **55.9**      | 55.8      |
>
> *Table B: The comparison of model ensemble experiments between CowMix and ClassMix. The number (e.g., 1) denotes different runs of the same DB method.*
> |           | CowMix1 | CowMix2 | ClassMix1 | ClassMix2 |
> |-----------|---------|---------|-----------|-----------|
> | CowMix1   | 50.6    | -       | -         | -         |
> | CowMix2   | 51.5    | 50.3    | -         | -         |
> | ClassMix1 | 55.2    | 54.8    | 55.5      | -         |
> | ClassMix2 | 55.6    | 55.4    | **55.9**      | 55.8      |
>
> **Q4: Supplemental experiments about FDA[A].**
>
> **A4:** This frequency domain-based style transfer technique belongs to the style transfer-based DB methods, in which $\beta$ controls the extent to which the amplitude spectrum is exchanged. Table C shows the results of source-only and pseudo-labeling adaptation experiments using the FDA technique, which we found to be more effective than those in Table 1(a). Additionally, as shown in Table D, we combined ClassMix and FDA(S→T) in the same way as Table 1(c) to conduct the model ensemble experiments and observed hardly any improvement.
>
> *Table C: The comparison of FDA under source-only (S→T) and pseudo-labeling (T→S) settings and various $\beta$.*
> |     | $\beta$ | mIoU |
> |-----|---------|------|
> | S→T | 0.01    | 40.9 |
> | S→T | 0.05    | **42.0** |
> | S→T | 0.09    | 41.2 |
> | T→S | 0.01    | 26.6 |
> | T→S | 0.05    | 39.3 |
> | T→S | 0.09    | **43.1** |
>
> *Table D: The comparison of the model ensemble combining ClassMix and FDA.*
> |                            | ClassMix+FDA($\beta=0.01$) | ClassMix+FDA($\beta=0.05$) | ClassMix+FDA($\beta=0.09$) |
> |----------------------------|----------------------------|----------------------------|----------------------------|
> | ClassMix+FDA($\beta=0.01$) | 52.7                       | -                          | -                          |
> | ClassMix+FDA($\beta=0.05$) | **52.9**                       | 52.4                       | -                          |
> | ClassMix+FDA($\beta=0.09$) | 52.4                       | 51.9                       | 50.2                       |

---

> ### Author Response · Authors · 2022-08-08
> **Further Discussion with Reviewer PF8F**
>
> Dear reviewer PF8F:
>
> We sincerely thank you for the time and comments. We have provided corresponding responses and experimental results, which we believe have covered your concerns. We hope to further discuss with you whether or not your concerns have been addressed. Please let us know if you still have any unclear parts of our work.
>
> Best,
> Authors of Paper 726

---

> > ### Comment · Reviewer_PF8F · 2022-08-09
> > **Appreciation of author feedback**
> >
> > Dear authors of Paper 726,
> >
> > I appreciate the detailed feedback and the additional experimental results. I think most of my questions have been adequately addressed, which makes me more confident in suggesting to accept the paper.

---

> > > ### Author Response · Authors · 2022-08-09
> > > **Author Response**
> > >
> > > We thank you again for your valuable comments and the kind support of this work.

---

### Official Review · Reviewer_u4Xd · 2022-07-10

**Rating:** 6
**Confidence:** 5
**Soundness:** 3 good
**Presentation:** 2 fair
**Contribution:** 2 fair

**Summary:**

This paper proposes a deliberated domain bridging (DDB) method for domain adaptative semantic segmentation, where the target labels are not available during the training. In DDB, there are two parts: 1) a dual-path domain bridging step to train two teacher models with two intermediated domains using the coarse-wise and fine-wise, i.e., region-level and semantic-level, data mixing techniques. 2) a cross-path knowledge distillation step to adaptively transfer the knowledge from the two teacher models to a student model. The two steps are repeated for several rounds for a good performance. Extensive experiments on both single-source domain and multi-source multi-target domain settings are conducted to validate DDB’s superiority.

**Questions:**

See the weakness.

**Limitations:**

The authors discuss the limitations of this paper in the supplementary material. No negative social impact has been discussed.

**Strengths And Weaknesses:**

Pros:
1. This paper proposes an effective method to significantly boost the UDA segmentation performance in various settings.
2. The comprehensive ablations are done to clearly show 1) the complementarity between the two teacher models and 2) the effectiveness of the distillation step.

Cons:
1. Since GTA5 to Cityscapes and GTA5 + Synscapes to Cityscapes are done, what is the performance in Synscapes to Cityscapes? This experiment shows which dataset contributes more to adapt to the real dataset.
2. There are too many symbols, which makes the paper hard to follow. What do the numbers righter after the approach name in Tables 2, 3, and 4 mean? For example, ADVENT(19), BDL (19), FADA(20), etc.
3. The authors claim that soft distillation and hard distillation are compared in Table 5. However, the ‘soft distillation’ choice and the explanation are missing in that table, which is a bit confusing.
4. DDB requires two rounds for a good convergence. In each round, it needs to train three individual models and calculate two groups of category centroids by scanning the target training set for two teacher models respectively. This makes the approach cumbersome and may require more training time than others. The authors are encouraged to discuss the above issue with detailed analysis. Besides, this cumbersome training process seems in conflict with the stated ‘elegant’ method.
5. The following paper can be included for comparison since it also studies the data mixing technique in UDA semantic segmentation. Besides, the difference between DACS which also utilizes the data mixing technique in UDA is not well stated in the paper.

Dsp: Dual soft-paste for unsupervised domain adaptive semantic segmentation. Proceedings of the 29th ACM International Conference on Multimedia. 2021: 2825-2833.

---

> ### Author Response · Authors · 2022-08-02
> **Response to Reviewer u4Xd**
>
> Thank you so much for the detailed and thoughtful review! We have fully addressed the concerns as follows:
>
> **Q1: The performance in Synscapes to Cityscapes.**
>
> **A1:** We further conducted experiments on the benchmark of Synscapes to Cityscapes and reported them below. An interesting finding is that even though the source-only models trained on the Synscapes dataset perform significantly better than those trained on GTA, the final adaptation performance of Synscapes is lower than that of GTA. We analyze this is because of the lack of fine-tuning of hyperparameters and the lack of diversity in the style of Synscapes. Moreover, our proposed framework not only achieves satisfactory adaptive performance on both GTA and Synscapes datasets but also their combination (GTA and Synscapes) can further improve the performance to an impressive 69.0% mIoU. This is thanks to our naturally constructing intermediate domains between target and multiple source domains to facilitate knowledge transfer.
> | Method      | mIoU     |
> |-------------|----------|
> | Source only | 47.8±0.2 |
> | Stage1 CRP  | 55.3±0.4 |
> | Stage1 FCP  | 55.9±0.7 |
> | Stage1 CKD  | **57.1**±0.9 |
>
> **Q2: What do the numbers righter after the approach name in Tables 2, 3, and 4 mean?**
>
> **A2:** This is an indication of the year in which the method was proposed, in order to make it easier for the reader to understand the currency of the method being compared. We have already updated the complete name for venues in the revision for clarity.
>
> **Q3: The ‘soft distillation’ choice and the explanation are missing in that table, which is a bit confusing.**
>
> **A3:** Thanks for your valuable suggestion to improve our paper. The column 'hard distillation' in the CKD step indicates that the hard distillation was used (if it is ticked), otherwise, it means soft distillation. For easy understanding, we have already updated the caption of this table in the revision.
>
> **Q4: Clarification about the training efficiency.**
>
> **A4:** We provided a detailed comparison of the training efficiency. As shown in the following table, our method achieved better performance after one round of training with smaller input size, fewer GPUs, fewer iterations, and fewer post-processing steps than other SOTA methods. Furthermore, our method could still achieve gains of 1.5% mIoU with one more training round, while still being more efficient than other SOTA methods.
> | Method             | Input size | Total GPUs | Total iterations | Pseudo-label generation | Prototype generation | mIoU |
> |--------------------|------------|------------|------------------|-------------------------|----------------------|------|
> | ProDA+*distill*[62]  | (896,512)  | 4          | 238K             | √                       | √                    | 57.5 |
> | UndoDA+*distill*[31] | (896,512)  | 4          | 238K             | √                       | √                    | 59.3 |
> | CPSL+*distill*[27]   | (896,512)  | 4          | 238K             | √                       | √                    | 60.8 |
> | DDB-round1        | (512,512)  | 1          | 80K              | ×                       | √                    | 61.2 |
> | DDB-round2         | (512,512)  | 1          | 160K             | ×                       | √                    | **62.7** |
>
> **Q5: The difference between DACS and DSP which also utilizes the data mixing technique in UDA is not well stated in the paper.**
>
> **A5:** We thank the reviewer for the valuable comment, and we have incorporated the missing references and discussed them as follows: Based on the cross-domain ClassMix technique, DACS only mixes the source and target domains to obtain an intermediate domain, but DSP mixes not only the source and target domains but also the source and source domains, so as to effectively reduce the domain gap. Different from them, we explored a variety of ways to construct DBs and found that the coarse-wise DB (e.g., CutMix) and fine-wise DB (e.g., ClassMix) methods were complementary. Based on this, we propose a dual-path DB method to effectively exploit such complementary properties to achieve better results.

---

> > ### Comment · Reviewer_u4Xd · 2022-08-08
> > **Post-rebuttal discussion**
> >
> > Thank the authors for the response.
> >
> > The authors conduct the Synscapes->Cityscapes adaptation experiments and provide the experiments. Despite only one round of training, i.e., one stage, which is less than the default setting, the improvement over the vanilla source-only result is clear and significant, showing the effectiveness of the proposed method.
> >
> > The authors also carefully discuss the reason why the improvement of Synscapes->Cityscapes is less than GTA5->Cityscapes, and give a table to compare the recent advanced methods w.r.t the training cost, where the results demonstrate that the proposed method can reach better performance with less training steps. It is highly encouraged to add this comparison to the paper.
> >
> > Besides, the authors also respond to the questions including the difference between the proposed method and DACS/DSP, the explanation in the table, more domain bridging strategies as pointed out by PF8F, the intermediate model selection as pointed out by i4CE, the ablation w.r.t. hyper-parameters by xZqY, etc.
> >
> > Overall, the major concerns have been addressed. Considering the novelty and technical contribution, I would like to keep my rating.

---

> > > ### Author Response · Authors · 2022-08-08
> > > **Author Response**
> > >
> > > Dear Reviewer u4Xd
> > >
> > > We sincerely thank you for the constructive feedback and support. And we promise to add the comparison table of the training cost to the supplementary material in the revision.

---

### Official Review · Reviewer_i4CE · 2022-07-10

**Rating:** 6
**Confidence:** 5
**Soundness:** 3 good
**Presentation:** 3 good
**Contribution:** 3 good

**Summary:**

To address UDA in semantic segmentation, this work uses two types of data mixing strategies to artificially create intermediate bridging domains between source and target. The paper starts with a detailed analysis comparing different data mixing strategies, either done globally (mixup[61]) or locally ( CowMix [13], FMix [17], CutMix [60] and ClassMix [42]). The analysis demonstrates favorable results when using local data mixing strategies for UDA in segmentation, in particular CutMix (coarse region-wise mixing) and ClassMix (fine class-wise mixing).

Based on results of the analysis, this work proposes a simple way to combine the two mixing strategies CutMix and ClassMix.
In the course of training, there are five models: two teacher models trained with CutMix and ClassMix, two EMA models of the two teachers, one student model trained using teachers' pseudo-labels.

Training is done in multiple rounds (fixed as 4 in the experiments). In each round:
- The two teachers are first trained separately with CutMix and ClassMix
- The student is then trained with pseudo-labels of two EMA models of the two teachers. Pseudo-label of a given target sample is determined as a weighted combination of softmax scores of the two EMA models (Eqn. 12). The weights have size $H \times W \ K$ with $K$ classes; at each spatial position, the weight vector over $K$ classes is the softmax over the feature distance to class centroids (Eqn. 11). Color jittering and gaussian blurs are used on target sample when training the student.
- The two teachers are initialized by the student.



**Questions:**

- DAFormer (CVPR'22) is a recent work demonstrating great DA performance thanks to the robustness of transformer-based Segformer model. As the proposed strategy is agnostic to the model choice, I'm wondering to which extent one can push the performance by combining the proposed and DAFormer

**Limitations:**

Limitation on training complexity is discussed in the supplementary material.
No discussion on potential negative societal impacts was given.

**Strengths And Weaknesses:**

** Strengths **
Overall this is an interesting technical paper that combines multiple existing strategies, namely CutMix [60], ClassMix [42], mean teacher [42], prototypical weighting [62]  and pseudo-labelling [30]. Empirical results demonstrate better performance than previous SOTAs on comparable backbone (resnet101) and segmentation framework (deeplab-v2). Experiments are extensive. The paper is well-written and easy to follow.

** Weaknesses **
- My main concern is with the technical novelties of this work. The analysis comparing different mixing techniques, claimed as the first contribution, is somewhat interesting. However the main proposed approach is merely a mix of previous works. Actually there are no new insights that I could get from this work.

- It's not clear to me how is the intermediate model selected at each stage. Is the target's validation set used to select the best model? If true, is there a risk of supervision leak from target validation set?

- Missing details for the multi-source and multi-target experiments.

I'm currently on the borderline, slightly leaning toward the positive side, thanks to the good results. My final decision will be adjusted based on the feedback from the authors and the discussion with other reviewers.

** Typos **
- L185: Eqn. 3 instead of Eqn. 2
- SuppMat: Algo.1 - L7 & L12: Eqn. 3 instead of Eqn. 2

========== Post-rebuttal
I thank the authors for being active during the rebuttal and addressing all of my concerns. I'm happy to increase my score.

---

> ### Author Response · Authors · 2022-08-02
> **Response to Reviewer i4CE**
>
> We are encouraged to see that you found our work interesting, extensive experiments contained, and well-written. We have endeavored to address your concerns as follows:
>
> **Q1: Clarification about the student model training.**
>
> **A1:** There seems to be a misunderstanding about the generation of pseudo labels in the CKD step. Notably, we use pseudo labels generated from two original teacher models rather than from the EMA models to teach the student model.
>
> **Q2: Clarification about the main contribution of this paper.**
>
> **A2:** First and foremost, we would like to thank you for acknowledging that our first contribution of analyzing different data mixing techniques is of interest. Then, we must emphasize that this work mainly aims to tackle the DASS task through a new perspective of domain bridging. Specifically, this work goes beyond a naïve/simple combination of existing mixings, like CutMix, ClassMix, etc., and innovatively explores a framework dubbed Deliberated Domain Bridging (DDB) to well-coordinate coarse region mixing (e.g., CutMix) and fine class mixing (e.g., ClassMix). Two independent bridging paths in Dual-path Domain Bridging (DPDB) are developed and focus on extracting complementary knowledge. In addition, a Cross-path Knowledge Distillation (CKD) step based on the teacher-student mechanism as well as a win-win alternating optimization strategy are carefully designed, which is the icing on the cake.
>
> **Q3: How the intermediate model is selected at each stage?**
>
> **A3:** We just use the latest checkpoint in each stage, which does not involve any supervision leak risk.
>
> **Q4: Details about the multi-source and multi-target experiments.**
>
> **A4:** In fact, we treat the multi-source datasets and multi-target datasets as one domain and **combine them directly**, which results in a single-source and single-target domain setting. Notably, there is **no tailored design** under these settings and the proposed DDB outperforms the previous SOTA methods by a large margin (10.0% and 11.1% mIoU gains under the multi-source and multi-target domain settings, respectively).
>
> **Q5: Typos about the reference of Eqn.3 instead of Eqn.2.**
>
> **A5:** We have already corrected these typos in the revision.
>
> Your constructive comments and criticisms will greatly assist us in improving this work. Please do not hesitate to contact us if you have any further questions.

---

> ### Author Response · Authors · 2022-08-08
> **Further Discussion with Reviewer i4CE**
>
> Dear reviewer i4CE:
>
> We sincerely thank you for the review and comments. We have provided corresponding responses and clarifications, which we believe have covered your concerns. We hope to further discuss with you whether or not your concerns have been addressed. Please let us know if you still have any unclear parts of our work.
>
> Best,
> Authors of Paper 726

---

> > ### Comment · Reviewer_i4CE · 2022-08-09
> > **All concerns are cleared**
> >
> > Dear authors, your response has cleared my concerns.
> > Thanks,
> > i4CE

---

> > > ### Author Response · Authors · 2022-08-09
> > > **Thanks for the appreciation**
> > >
> > > Thanks very much for your appreciation and recognition，but may I respectively ask you to rise your rating for our paper to promise an acceptance, thanks.

---

### Official Review · Reviewer_XZqY · 2022-07-12

**Rating:** 6
**Confidence:** 3
**Soundness:** 3 good
**Presentation:** 3 good
**Contribution:** 3 good

**Summary:**

This paper proposes an effective Deliberated Domain Bridging (DDB) for domain adaptive semantic segmentation (DASS). To this end, it takes advantage of two data mixing techniques, region-level mix and class-level mix, to train two corresponding teacher models, which eventually guide one student model on the target domain. It has been tested on several benchmarks (GTA5 to Cityscapes, GTA5 + Synscapes to Cityscapes, GTA5 to Cityscapes + Mapillary).


**Questions:**

Please answer the questions from the above weaknesses.

**Limitations:**

Yes, the authors addressed the limitation.

**Strengths And Weaknesses:**

- Strengths:
1. It is a well-written paper that addresses the limitations of previous methods (e.g., global interpolation -> pixel-wise ambiguity), toy game to show the justification of using both coarse-grained and fine-grained DB, and proposes novel learning architecture with multi-teacher and single-source distillation method.
2. The proposed method is well proven to be effective with several benchmarks (e.g., single-source, multi-source, and multi-target settings) by widening the gap with the previous state of the arts in each benchmark.

- Weaknesses:
1. As the author illustrated in limitations, I am also a bit concerned with the training efficiency and complexity since the proposed method requires alternating optimization processes. One of the simple end-to-end optimizations is conducting EMA training on one model by combining region-level and class-level mixing techniques. It would be better to show a brief study about how the authors can extend this to end-to-end simple learning architecture.
2. Lack of ablation study on some hyperparameters: 1) alpha in equation 5: the authors suggested updating the teacher models with EMA to avoid a denoised pixel-wise pseudo label. To prove it, an ablation study on alpha needs to be explored. 2) x_aug in equation 9: need to empirically show justification using augmentation input for a student model.

---

> ### Author Response · Authors · 2022-08-02
> **Response to Reviewer xZqY**
>
> We thank you for the positive comments on the novelty of our approach and paper writing. We also appreciate your valuable suggestions on the supplement of ablation experiments. We have accordingly refined our paper as follows:
>
> **Q1: About the training efficiency and complexity.**
>
> **A1:** Our method is more efficient than other SOTA methods, compared to recent methods such as ProDA [62], UndoDA [31], and CPSL [27] where they typically use **four** GPUs and a total of **four** training rounds (about 60K iterations in each round, and one round of warm-up, one round of self-training, and two rounds of self-supervised self-distillation), our method only uses  **one** GPU after performing **one** round of DPDB and CKD steps (40K iterations in each step) but achieves a superior performance of 61.2% mIoU. Moreover, these compared methods have almost reached the performance bottleneck after four rounds of training, while our method could still achieves gains of 1.5% mIoU in the second round of optimization.
>
> **Q2: Show a brief study about how the authors can extend the proposed approach to end-to-end simple learning architecture.**
>
> **A2:** Thanks. We have accordingly tried a heuristic end-to-end learning architecture as follows:
> * ***Architecture:*** Instead of leveraging two independent teacher models to learn complementary knowledge in coarse region-path and fine class-path, we propose a single model equipped with a shared backbone and two segmentation heads for distinct DB paths and one attention head for fusing the segmentation predictions. Specifically, the attention head has the same architecture as segmentation heads, while the last convolution layer only has one output channel and is followed by a Sigmoid activation function. Moreover, we build an EMA model for generating the denoised pseudo label on the fly.
> *  ***Pipeline:*** We first feed the source image into the end-to-end model to calculate the cross-entropy loss items in the source domain like Eqn. 4. Next, we utilize the fused prediction of the EMA model to generate the pseudo label and confidence-based weight map for the clean target image, which enables the calculation of bridging loss items like Eqn. 7 for different bridging images and their corresponding segmentation heads. Moreover, we feed the augmented target image into the model and calculate the cross-entropy loss on the fused prediction and pseudo label for optimizing the attention head.
> * ***Results:*** We conducted experiments on the GTA5→Cityscapes benchmark and listed the performance of the end-to-end scheme and the original scheme. As shown in the table below, the results of the end-to-end manner in the single-stage are better than the DPDB even with the average ensemble. Moreover, as a result of the end-to-end architecture integrating complementary knowledge from the two domain bridging paths in the training process, both CRP and FCP variants perform better than the original ones.
> | Method              | mIoU |
> |---------------------|------|
> | Stage1 CRP          | 56.5 |
> | Stage1 FCP          | 58.2 |
> | Stage1 Avg Ensemble | 59.3 |
> | End-to-End CRP      | 58.7 |
> | End-to-End FCP      | 59.3 |
> | End-to-End Fused    | **60.1** |
>
> **Q3: Ablation study on the momentum of EMA in the DPDB step.**
>
> **A3:** Thanks. We setted the momentum $\alpha$  following the most of the mainstream self-training DASS methods such as DACS [49], DAFormer [23], etc. Here, we further conducted ablation experiments on $\alpha$, and found that $\alpha=0.99$ will lead to better and more stable results.
> | Momentum | 0.9      | 0.99     | 0.999    | 0.9999   |
> |----------|----------|----------|----------|----------|
> | mIoU     | 58.3±0.4 | **58.5**±0.3 | 58.4±0.3 | 54.2±0.1 |
>
> **Q4: Ablation study on the augmentation strategy for the input of the student model in the CKD step.**
>
> **A4:** Thanks. We further conducted corresponding ablation experiments and reported the results. As shown below, combining the Gaussian blur and color jitter augmentation techniques leads to the best performance.
> | Gaussian blur | Color jitter | mIoU     |
> |---------------|--------------|----------|
> |               |              | 61.0±0.2 |
> | √             |              | 61.2±0.2 |
> | √             | √            | **61.5**±0.3 |

---

### Author Response · Authors · 2022-08-02
**General Response to All Reviewers**

We sincerely appreciate all reviewers for your time and efforts in the review. We are delighted and encouraged to see our paper get all positive ratings, and we have carefully rephrased our paper (see red marks) to correct typos and update missing citations in the revision. Other detailed questions are answered accordingly in each column below.

---

### Meta-Review · Area_Chair_LZAu · 2022-08-24

**Recommendation:** Accept
**Confidence:** Certain

**Metareview:**

**Summary**: This paper proposes an effective Deliberated Domain Bridging (DDB) approach for domain adaptive semantic segmentation (DASS). It leverages two data mixing techniques: region-level mix and class-level mix, to train two corresponding teacher models, which then guide one student model on the target domain. It is evaluated on multiple benchmarks.

**Strength**: The paper is a well-written paper. It is well-motivated based on the limitations of previous methods. The proposed approach is novel, interesting, and effective. The experiments (with the toy game) are solid.

**Weakness**: Training efficiency and complexity. Lack of ablation study on some hyperparameters and design choices. Some missing references/comparisons; unclear positioning of the work w.r.t. prior work.

**Recommendation**: The paper receives consistently positive ratings. After rebuttal, most of the reviewers’ concerns are addressed and the paper clearly has strengths. The AC thus suggests acceptance. The AC strongly suggests that the authors incorporate their rebuttal (e.g., additional results) into their camera-ready version.


**Award:**

No

---

### Decision · Program_Chairs · 2022-09-14

Accept